# The Use of Electrode Tools Obtained by Selective Laser Melting to Create Textured Surfaces

**DOI:** 10.3390/ma15144885

**Published:** 2022-07-13

**Authors:** Timur Rizovich Ablyaz, Evgeny Sergeevich Shlykov, Karim Ravilevich Muratov

**Affiliations:** Department of Mechanical Engineering, Perm National Research Polytechnic University, 614000 Perm, Russia; kruspert@mail.ru (E.S.S.); karimur_80@mail.ru (K.R.M.)

**Keywords:** texture, oil-retaining grooves, complex profile relief, electrical discharge machining, electrode tool, selective laser melting technology, additive manufacturing

## Abstract

The study and development of the technological foundations for creating a textured surface using an electrode tool obtained by the method of additive manufacturing are the purpose of the work. Methods for obtaining textured surfaces and for creating a tool electrode for electrical discharge machining are considered in this work. The modeling of the electrodetool, analysis of internal stresses during its manufacture by the selective laser melting method, and the manufacture of electrodes are considered. A Realizer SLM 50 laser machine was used to create the electrode tool. Ti6Al4V metal powder with an average particle size of 30 µm was chosen as the material for manufacturing. The experiments were carried out on a copy-piercing electrical discharge Smart CNC machine. The material of the workpiece is corrosion-resistant, heat-resistant, high-alloy steel 15Cr12H2MoWVNNB. An Olympus GX 51 light microscope (Olympus Corporation, Shinjuku-ku, Japan) at 100× magnification was used to visually evaluate the texturing results and measure dimensions. The possibility of using electrodes obtained by the selective laser melting method for texturing surfaces was studied.

## 1. Introduction

An urgent task of mechanical engineering is to improve the quality of manufacturing high-precision and critical parts with an applied textured profile. The complex profile of parts and the use of modern structural materials with improved physical and mechanical properties do not allow the full use of traditional processing methods.

In works [1,2,3,4,5,6], it was noted that in order to improve the operational characteristics of the working surfaces of products, for example, working in conditions of increased friction, it is necessary to apply a textured relief. A textured relief is applied to individual surfaces requiring certain properties. Surface texturing is a method of surface functionalization widely used throughout the world. This is used in various fields: medical implants, wettability tuning, optical properties, hybrid bonding, bond enhancement, or cutting tools [1,2,3,4]. The most common patterns of textured surfaces are dimples, lines (parallel and hatching), square, conical, rhombic, wavy texture, and annular (bulge and smooth corrugation) [3,4]. Surface microtexturing is widely used to improve the tribological properties of cutting tools, such as improving frictional performance and wear protection. The types of patterns produced on the surfaces of the cutting tool were most commonly microgrooves, microholes, microstrip grooves, and microgrooves. In the works [1,2,3,4,5], the influence of polycrystalline diamond tools with microgrooves, made using a fiber laser, was studied (Figure 1).

Textured surfaces are widely used in mechanical engineering. Theyare used to create finished products and mold dies and when creating a special auxiliary element in the form of oil-retaining surfaces in highly loaded friction pairs. The quality and reliability of the entire mechanism depend on the accuracy of applying the textured surface to the working surfaces of the products. It was noted in [5,6] that the application of a textured relief can improve the strength properties of products due to the creation of a fine-grained structure on the surface layer. In addition, the coefficient of friction is reduced by obtaining microstructured oil-retaining cavities. The shape and depth of the resulting pattern and complex relief are the main requirements for creating oil-retaining grooves on the surface of products [1,2,3,4].

The technological process of applying a special pattern to the surface of a precision part using the blade method is expensive and technologically inefficient. A special tool is used to create a complex profile. This tool has high wear during machining. High temperatures occur in the cutting zone. These factors lead to an increase in the cost of manufacturing a part and the frequent occurrence of defects in the production process.

Electrical discharge machining (EDM) technology is used to eliminate possible losses and improve the surface quality. This technology makes it possible to obtain the required surface profile without significant distortion.

The manufacture of an electrode tool (ET) is a limiting factor in the development of EDM technology. ETs made of copper, brass, bronze, and other nonferrous alloys are widely used in EDM. However, obtaining a complex profile on the ET surface is an economically inefficient process and an expensive process. This is due to the high cost and labor intensity of the technology. Most of the metal goes into chips during the manufacture of electrodes from these materials, which makes the process of their manufacture economically unprofitable [1].

A high pace of development of digital technologies in mechanical engineering is currently observed. Additive technologies are widely used in the creation of fixtures, dies, and tools. Their development is due to the fact that high quality requirements are imposed on finished products [1,2,3,4,5].

Selective laser melting (SLM) technology is one of the ways to obtain ETs. This consists of layer-by-layer melting of the metal powder to obtain the finished product [7,8,9,10]. This method makes it possible to obtain ETs practically without the use of mechanical operations. These operations are necessary in order to extract the constructed part from the substrate on which it is grown, and it is also possible to finish the surface with ETs by grinding [11,12].

The performance of the EDM ET and the applicability of the ET produced by the selective laser melting method for EDM have not been fully studied [13,14,15,16,17,18]. The development of EDM technology by grown ETs obtained by the SLM method for applying a textured surface is relevant [19,20] (Figure 2).

The study and development of the technological foundations for creating a textured surface using an electrode tool obtained by the method of additive manufacturing are the purpose of the work.

## 2. Materials and Methods

A Realizer SLM-50 laser machine was used to create ETs. The size of the zone for constructions was Ø70 × 35 mm. A pressure of 8 mbar was maintained in the working chamber. Argon was used as a protective atmosphere. A fiber laser was used in the process, with a power of 50 W and a wavelength of 680 nm. Titanium powder particles were sintered on a titanium substrate. The thickness of the treated layer was 30 μm.The metal powder Ti6Al4V Starbond Ti4 Powder 45, S&S Scheftner GmbH, Mainz, Germany, with an average particle size of 30 µm and a shape close to a sphere was chosen as the material for manufacturing.

The experiments were carried out on a copy-piercing electroerosive Smart CNC machine (Electronica India Limited, West Bengal, India). The material of the workpiece is corrosion-resistant, heat-resistant, high-alloy steel 15Cr12H2MoWVNNB. Mechanical processing (grinding) of the working end was carried out after removing the ET from the substrate in order to remove the supports.

Processing modes are presented in Table 1.

Processing was carried out in the machine tank filled with transformer oil. An Olympus GX 51 light microscope at 100× magnification was used to visually evaluate texturing results andto measure dimensions. Theoretical modeling was carried out using Siemens NX v.11, Materialize Magics v.22, and ANSYS Additive software 2019. The ETwas modeled in two types:ET with a square section and a side measuring 19 mm (Figure 3a) and a cylindrical ET with a diameter of 19 mm (Figure 3b). The end surface of both electrodes was made in the form of square cells with a cell size of 2 mm and a partition thickness of 1 mm. The height of the working surface of the ETwas 3 mm.

A 7 × 10 mm cylinder was provided for fixing the electrode in the electrode holder (Figure 4).

The process of preparing a 3D model for manufacturing included the following steps:Setting the working space of the installation in the CAD system (Siemens NX v.11) and placing the required number of elements on the platform;Construction of supports;Modeling the construction process in ANSYSAdditive;Correction of the model and supports (if necessary);Construction of the part.

The preparation of the model for manufacturing was carried out in the Materialize Magics environment. A cylinder with dimensions of 70 × 40 mm was the working space of the installation for making a model (Figure 5a).This allowed to reduce the time and reduce the cost of manufacturing. Supports were created in the minimum volume. This volume was sufficient to hold the grown product. (Figure 5b). The support type was a block. The support material entered the main part to a depth of 0.1–0.2 mm for reliable fastening of the part to the supports (Figure 5c).

After designing the supports, it was necessary to evaluate the internal stresses. These stresses arose during the construction process. An internal stress analysis was performed in the ANSYS Additive environment.

## 3. Results

### 3.1. Simulation of Internal Stresses

The forces arising during the construction process were applied to the part in the process of modeling the additive process. The part was deformed. Creating a preliminary deformation allows you to change the original model. When external loads are applied to solids, the bodies can be deformed up to destruction. The degree of deformation depends on the magnitude of the load. Bodies return to their original shape under small loads after they are removed. Since the deformed bodies are in equilibrium, the external loads must be balanced by internal forces. Deformation leads to a change in the crystal lattice. Inner forces appear. These forces are trying to return the system to its original state. As a result, forces begin to act on each elementary particle of the body. These forces together form the stress vector P.

To determine the stress vector P, it is necessary to introduce the concept of the stress tensor $\sigma_{ij}$ (Formula (1)).

We introduced the stress tensor as follows:(1)$$\sigma_{ij}=\left\{\begin{array}{ccc} \sigma_{11} & \sigma_{12} & \sigma_{13}\\ \sigma_{21} & \sigma_{22} & \sigma_{23}\\ \sigma_{31} & \sigma_{32} & \sigma_{33} \end{array}\right\}$$

For the element $\sigma_{ij}$, the first index shows on which site the given stress acts, and the second index shows in which direction. If the indices are the same, then this is a normal stress, and if they are different, it is a shear stress.

If the stress tensor is known, then the stress vector (Formula (2)) can be defined as:(2)$$P_{nj}=\sigma_{ij}\cdot n_{i}$$
(3)$$\begin{array}{c} P_{n1}=\sigma_{11}\cdot n_{1}+\sigma_{12}\cdot n_{2}+\sigma_{13}\cdot n_{3}\\ P_{n2}=\sigma_{21}\cdot n_{1}+\sigma_{22}\cdot n_{2}+\sigma_{23}\cdot n_{3}\\ P_{n3}=\sigma_{31}\cdot n_{1}+\sigma_{32}\cdot n_{2}+\sigma_{33}\cdot n_{3} \end{array}$$
(4)$$\vec{P}=\left\{P_{n1},P_{n2},P_{n3}\right\}$$
(5)$$\vec{n}=\left\{n_{1},n_{2},n_{3}\right\}$$

If the stress tensor is determined in a given coordinate system, then, in the future, it is always possible to determine the stress vector (internal forces) acting on any arbitrarily chosen area.

The deformable body, as a whole, will be in equilibrium if the main vector of forces $\vec{R}$ and the main vector of moments $\vec{M}$ of mass and surface forces are equal to zero.
(6)$$\vec{R}=\overset{}{\underset{S}{\iint }}\vec{t}dt+\overset{}{\underset{V}{∭}}\vec{f}\cdot \rho dv=0$$
(7)$$\vec{M}=\overset{}{\underset{S}{\iint }}(\vec{\tau }\times \vec{t)}dS+\overset{}{\underset{V}{∭}}(\vec{\tau }\times \vec{f)}\rho dV=0$$
where $\vec{\tau }$ is radius vector of an arbitrary point;

$\vec{t}$ is force;

$\vec{f}$ is body forces;

ds is elementary section;

dv is elementary volume;

ρ isdensity of the material.

Select an arbitrary region V* defined by the surface S*.

On the surface S*, forces are distributed, which are internal forces for the volume V and external for the volume V*. Therefore, if the internal volume is in equilibrium, then:(8)$$\vec{R}=\overset{}{\underset{S^{*}}{\iint }}\vec{P_{n}}dS+\overset{}{\underset{V^{*}}{∭}}\vec{f}\cdot \rho dV=0$$
(9)$$\vec{M}=\overset{}{\underset{S^{*}}{\iint }}(\vec{\tau }\times \vec{P_{n}})dS+\overset{}{\underset{V^{*}}{∭}}(\vec{\tau }\times \vec{f})\rho dV=0$$

From the Ostrogradsky–Gauss formula it follows:(10)$$\overset{}{\underset{S^{*}}{\iint }}dS=\overset{}{\underset{S^{*}}{\iint }}\vec{P_{i}}\cdot n_{i}dS=\overset{}{\underset{V^{*}}{∭}}\frac{\partial P_{i}}{\partial x_{i}}dV$$

From Formula (10) it follows:(11)$$\overset{}{\underset{V^{*}}{∭}}\left(\frac{\partial P_{i}}{\partial x_{i}}+\rho \vec{f}\right)dV=0$$

Since V* is an arbitrary volume, equality is possible when
(12)$$\frac{\partial P_{i}}{\partial x_{i}}+\rho \vec{f}=\frac{\partial \sigma_{ij}}{\partial x_{i}}+\rho \vec{f}=0$$

Equation (12) is an equilibrium equation in differential form.

By analogy with Newton’s 2nd law (based on the D’Alembert principle), we can obtain the equation of motion in the form:(13)$$\vec{m}a=\sum F_{i}\to \frac{\partial \sigma_{ij}}{\partial x_{i}}+\rho \vec{f}=\rho \frac{\partial^{2}u_{i}}{\partial t^{2}}$$

Each term in Equation (13) has its own meaning.

$\frac{\partial^{2}u_{i}}{\partial t^{2}}$ is forces of inertia in some elementary volume;

$\rho \vec{f}$ is body forces (body forces);

$\frac{\partial \sigma_{ij}}{\partial x_{i}}$ is internal forces (a term that takes into account internal forces in the body).

The strain tensor is a tensor that characterizes the tension (compression) and change in the shape of each point of the body during deformation.

In a classical continuous medium, in which particles are material points and have only three translational displacements, but no rotation, the strain tensor (the Cauchy–Green tensor) is generally defined as:(14)$$\varepsilon_{j}=\frac{1}{2}\left(\frac{\partial u_{i}}{\partial x_{j}}+\frac{\partial u_{j}}{\partial x_{i}}+\underset{l}{\sum }\frac{\partial u_{l}}{\partial x_{i}}\cdot \frac{\partial u_{l}}{\partial x_{j}}\right)$$
where $\bar{u}=\left\{u_{1},u_{2},u_{3}\right\}$ is a vector describing the displacement of a body point.

The third term in the strain tensor (14) makes it possible to correctly take into account complex deformation in the case of large deformations. However, the presence of this term significantly complicates the solution of the problem of mechanics; therefore, in the case of small deformations (ε << 1), this term can be neglected, writing the strain tensor in the form:(15)$$\varepsilon_{j}=\frac{1}{2}\left(\frac{\partial u_{i}}{\partial x_{j}}+\frac{\partial u_{j}}{\partial x_{i}}\right)or\varepsilon_{ij}=\frac{1}{2}\left(u_{i,j}+u_{j,i}\right)$$

The connection between displacements and deformations is called geometric relations.

In the Cartesian coordinate system, geometric relationships can be expressed:(16)$$\bar{u}=\left\{u_{x},u_{y},u_{z}\right\}$$
(17)$${\hat{\varepsilon }}_{xx}=\varepsilon_{x}=\left[\begin{array}{ccc} \varepsilon_{x} & \varepsilon_{xy} & \varepsilon_{xz}\\ \varepsilon_{yx} & \varepsilon_{y} & \varepsilon_{yz}\\ \varepsilon_{zx} & \varepsilon_{zy} & \varepsilon_{z} \end{array}\right]$$
(18)$$\varepsilon_{x}=\frac{1}{2}\left(\frac{\partial u_{x}}{\partial x}+\frac{\partial u_{x}}{\partial x}\right)=\frac{\partial u_{x}}{\partial x}$$
(19)$$\varepsilon_{y}=\frac{1}{2}\left(\frac{\partial u_{y}}{\partial y}+\frac{\partial u_{y}}{\partial y}\right)=\frac{\partial u_{y}}{\partial y}$$
(20)$$\varepsilon_{z}=\frac{1}{2}\left(\frac{\partial u_{z}}{\partial z}+\frac{\partial u_{z}}{\partial z}\right)=\frac{\partial u_{z}}{\partial z}$$
(21)$$\varepsilon_{xy}=\frac{1}{2}\left(\frac{\partial u_{x}}{\partial y}+\frac{\partial u_{y}}{\partial x}\right)$$
(22)$$\varepsilon_{xz}=\frac{1}{2}\left(\frac{\partial u_{x}}{\partial z}+\frac{\partial u_{z}}{\partial x}\right)$$
(23)$$\varepsilon_{yz}=\frac{1}{2}\left(\frac{\partial u_{y}}{\partial z}+\frac{\partial u_{z}}{\partial y}\right)$$

Physical relationships are the relationship between stresses and strains. Many physical relationships are nonlinear dependencies, which are described by differential or integrodifferential equations. 

In general, the physical relationships look like:(24)$$\hat{\sigma }=f\left(\hat{\varepsilon }\right)$$

Generalized Hooke’s law:(25)$$\sigma_{ij}=C_{ijkl}\cdot \varepsilon_{kl}$$

The factor $C_{ijkl}$ includes 81 parameters, but for a linear isotropic material, two characteristics remain: Young’s modulus (E) and Punch’s ratio (ε). For elastic materials, Hook’s law is expressed by Formula (26):(26)σ=E·ε

For a linear, isotropic material, Hooke’s law is expressed by (27):(27)$$\sigma_{ij}={\lambda \Theta \delta }_{ij}+2\mu \varepsilon_{ij}$$
where λ and *μ* are Lame’s parameters;

$\Theta =\varepsilon_{x}+\varepsilon_{y}+\varepsilon_{z}$ is the volumetric strain;

$\delta_{ij}=\left[\begin{array}{c} 1,\;i=j\\ 0,\;i\neq j \end{array}\right]$ is the Kronecker symbol.

Based on Equations (25) and (27), normal and shear stresses can be determined:(28)$$\sigma_{x}=\lambda \Theta +2\mu \varepsilon_{x}$$
(29)$$\sigma_{y}=\lambda \Theta +2\mu \varepsilon_{y}$$
(30)$$\sigma_{z}=\lambda \Theta +2\mu \varepsilon_{z}$$
(31)$$\sigma_{xy}=2\mu \varepsilon_{xy}$$
(32)$$\sigma_{xz}=2\mu \varepsilon_{xz}$$
(33)$$\sigma_{yz}=2\mu \varepsilon_{yz}$$

Instead of the Lame’s parameters, other technical parameters (material characteristics) often determined from the experiment can be used.

Further, the temperature deformations and stresses during heating are taken into account. Let the body heat up by ΔT from temperature T_1_ to T_2_, i.e.,T_2_ = T_1_+ ΔT. Then, the total deformations can be written:(34)$$\varepsilon_{\mathrm{total}}=\varepsilon_{\mathrm{elast}}+\varepsilon_{\mathrm{temperature}}$$

Temperature deformations can be expressed in the form (35):(35)$$\varepsilon_{ij}^{T}=\alpha \Delta T\delta_{ij}$$
where α is the linear coefficient of thermal expansion.

Then, elastic deformations can be expressed in the form (36):(36)$$\varepsilon_{ij}^{\mathrm{elast}}=\varepsilon_{ij}^{\mathrm{total}}-\varepsilon_{ij}^{T}=\varepsilon_{ij}^{\mathrm{total}}-\alpha \Delta T\delta_{ij}$$
where $\varepsilon_{ij}^{elast}$ is elastic deformations;

$\varepsilon_{ij}^{total}$ is total deformations (determined from geometric relationships);

$\varepsilon_{ij}^{T}$ is temperature deformations.

If we substitute Equation (36) into Hooke’s law, we obtain:(37)$$\sigma_{ij}={\lambda \Theta \delta }_{ij}+2\mu \varepsilon_{ij}-3K\alpha \Delta T\delta_{ij}$$
where $K=\lambda +\frac{2}{3}\mu$.

To perform the analysis, we set the parameters of the Ti6Al4V material: modulus of elasticity E = 116 GPa and Poisson’s ratio *μ* = 0.31. The obtained values of the deformation of the EI are shown in Figure 6.

As a result of calculations for internal stresses when constructing the ET, it was revealed that the largest displacements are observed in the corners of the electrode (1.6 mm) (Figure 6).

### 3.2. Processing Grown ETs

An ET with a square base andan ET with a round base are obtained as a result of modeling and manufacturing using the SLM technology.

The wear of an ET is estimated after processing the surface of the workpiece. This is shown in Table 2. The processing was carried out on a round and square section of the ET on two samples.

It is shown that the round ET showed minimal wear of the ET. The amount of wear with a round ET is more than two times less than when machining with a square ET.

The creation of a special textured pattern on the surface was carried out using a cellular ET with different cell parameters (Figure 7). The relief was formed as a result of processing on the surface of the workpiece. This figure repeats the end surface of the ET [21,22,23,24,25].

It has been established that the cell profile has a different character (Figure 7 and Figure 8). This is due to the fact that the ET surface is formed by sintering a spherical powder. This powder does not provide 100% surface straightness. This factor indicates uneven processing and the absence of clear patterns in the construction of a textured relief at the microlevel. Figure 7 and Figure 8 shows the machined surfaces of the workpiece at 100× magnification.

Copying of the end face of the ET and traces of powder alloys are observed on the treated surface. Multilevel surface treatment is observed. When processing with a square ET, the size of the textured depressions is greater than when processing with a round ET (Figure 9).

On the basis of the experiment, it was shown that a textured surface with complex macrogeometry and microgeometry on a machined workpiece was created using the grown EE (SLM method). The presence of convex surfaces of fused powder material on the ET surface made it possible to form a textured system of multimodal roughness on the treated surface. The formation of tested surfaces by the EDM method using the grown ETon products operating under frictional interaction conditions increases their oiliness and, accordingly, hydrophobicity.

## 4. Conclusions

The use of ET 3D printing methods has expanded the possibilities of electrical discharge machining. It becomes possible to create ETs for surface texturing. This is an important and separate direction for the development of EDM. This paper shows an important point using the example of oil-retaining grooves. It is shown that SLM electrodes make it possible to make not only a macrorelief, but also a microrelief of a textured surface.

A study of the technology for creating ETs by the SLM method for EDM was carried out. The use of additive technologies makes it possible to manufacture a complex-profile ET with the required end surface topography.Simulation of the ET manufacturing process was carried out. A model that allows describing the manufacturing process and assessing the internal stresses that occur during construction, as well as a possible change in the shape and dimensions of the finished part, was obtained in the work. It has been established that during the manufacturing process of the part, the greatest internal stresses occur in the corners of the ET. These values allow us to make adjustments to the frequency of supports for additional fastening of the part to the machine substrate.The technology for creating a textured surface of the ET produced by the SLM method was developed. The creation of oil-retaining grooves on the surface of a product using EDM has a great practical application in mechanical engineering. As a result of technology development, oil-retaining grooves were created on the parts of the friction pair. These grooves improve the performance properties of the product.

## Figures and Tables

**Figure 1 materials-15-04885-f001:**
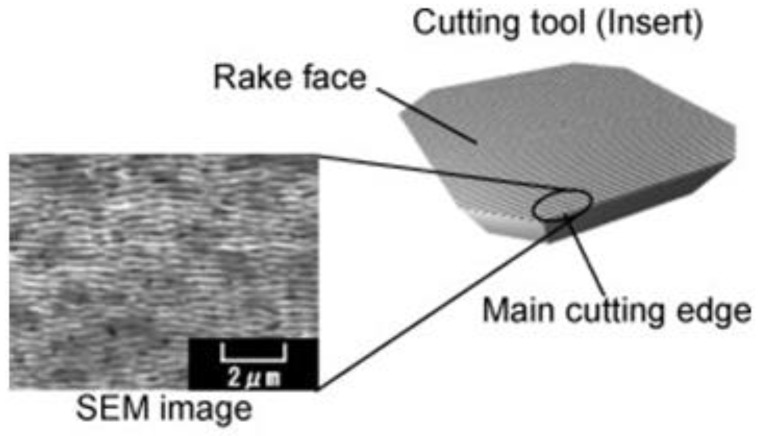
Surface of a textured cutting tool.

**Figure 2 materials-15-04885-f002:**
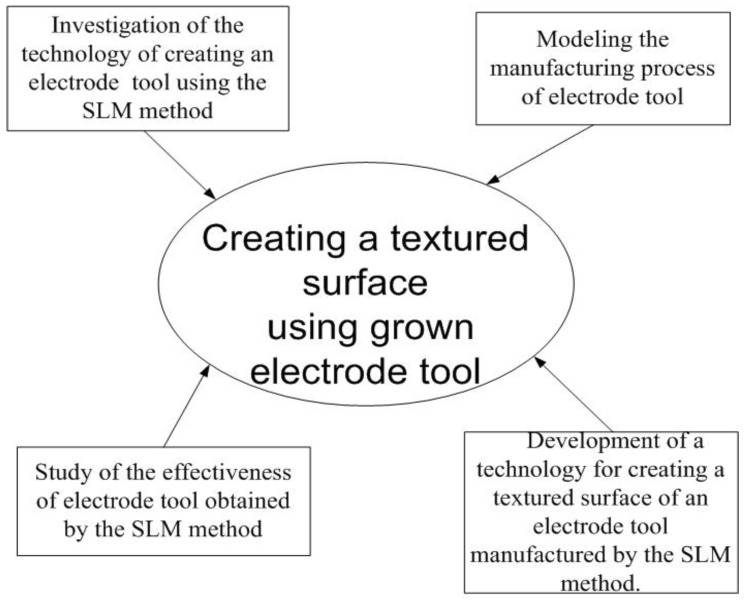
Stages of creating a textured surface using the grown ET.

**Figure 3 materials-15-04885-f003:**
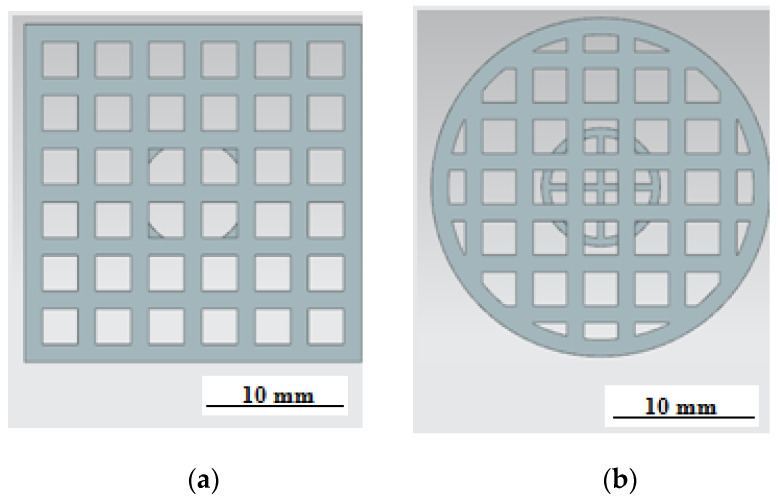
Three-dimensionalmodel of the electrode tool: (**a**) square ET and (**b**) round ET.

**Figure 4 materials-15-04885-f004:**
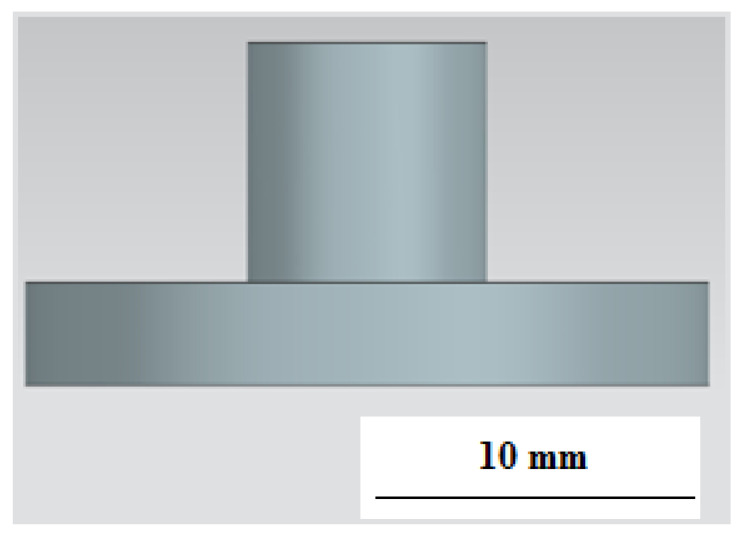
ET model (side view).

**Figure 5 materials-15-04885-f005:**
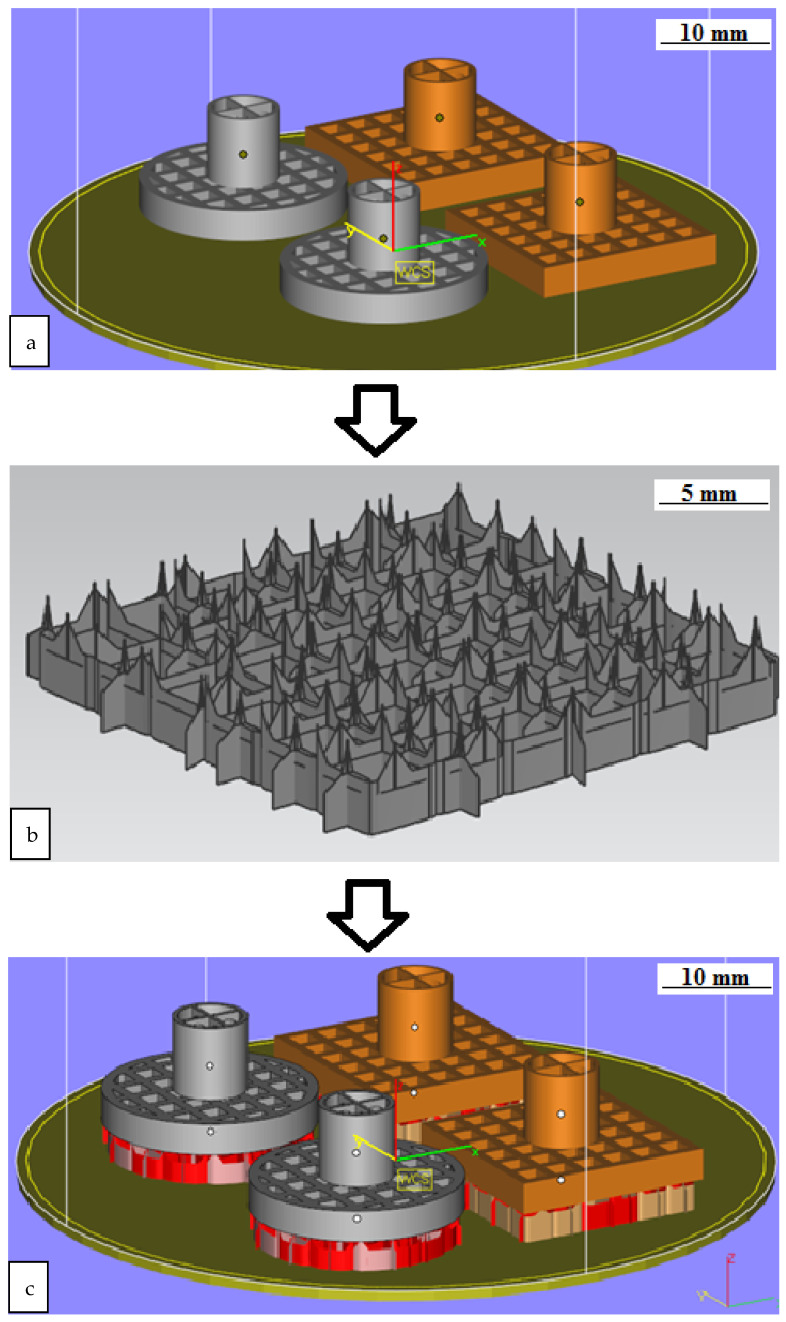
ETmodeling: (**a**) Materialize Magics workspace with placement of models for manufacturing; (**b**) supports for square ET; (**c**) arrangement of models on supports in the working space of the installation.

**Figure 6 materials-15-04885-f006:**
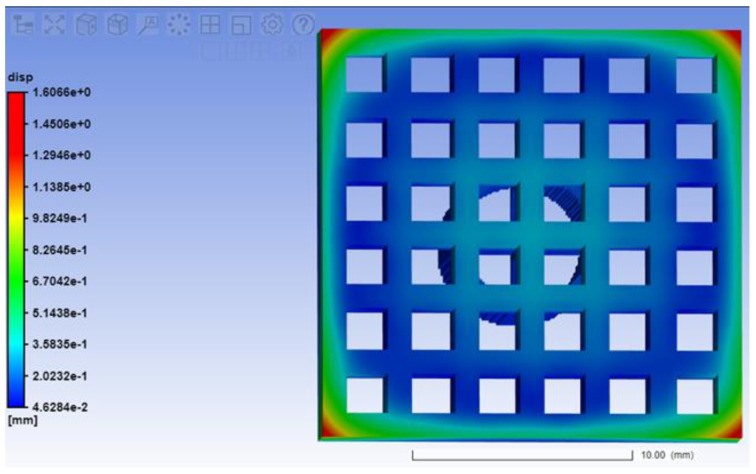
Deformation of the ET after separation from the supports.

**Figure 7 materials-15-04885-f007:**
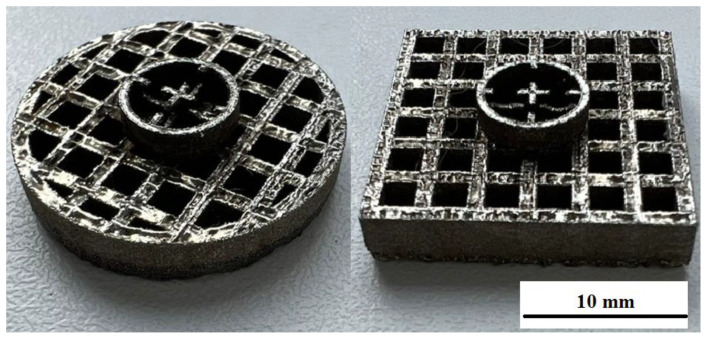
ET manufactured using SLM technology.

**Figure 8 materials-15-04885-f008:**
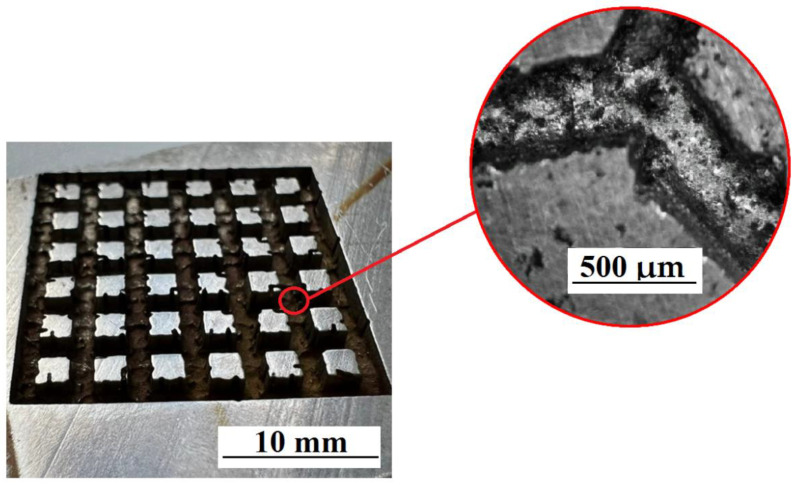
ET with a square section.

**Figure 9 materials-15-04885-f009:**
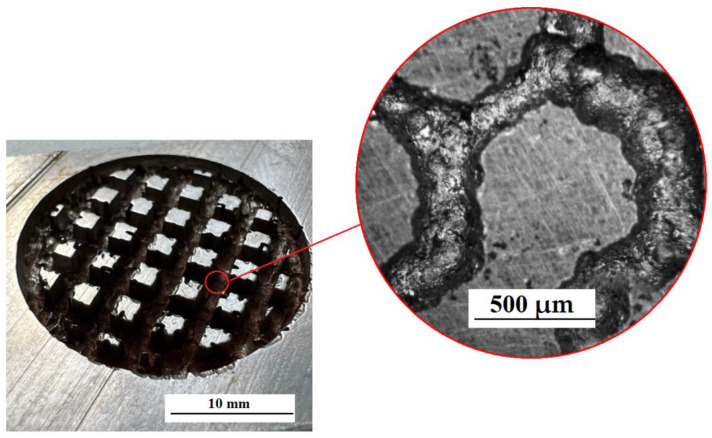
Treated surface ET with a circular cross section.

**Table 1 materials-15-04885-t001:** Processing mode.

Pulse on time (Ton), µs	100
Current (I), A	4
Voltage (U), V	50
Depth of processing, mm	2

**Table 2 materials-15-04885-t002:** The amount of wear of an ET.

Electrode No.	Average Height of the Working Part of the Electrode mm		Wear Rate, %
	New	After Processing	
1 (round)	35,795	35,723	0.2
2 (square)	35,202	35,026	0.5

## Data Availability

Not applicable.

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
