# Peer review of "The Use of Electrode Tools Obtained by Selective Laser Melting to Create Textured Surfaces"

_materials, 2022, doi:10.3390/ma15144885_

Round 1
Reviewer 1 Report
General comment:
This paper presents the study of electro discharge machining process used to create a textured surface. Part of the work is dedicated to analyze the influence of operating parameters (voltage, current and pulse duration) on cut speed, using regression dependence. Another part deals with the influence of the electrode tool geometry on EDM performances. Two configurations are considered, square and circular arrangement. The process of fabrication of the electrode tool using 3-D printing (selective laser melting) is also considered, in particular concerning the resulting internal stresses and deformation. While interesting results are presented it is difficult to link all parts. The diagram in Figure 1 presents all theses parts as individual tasks, without information on how they interact and if some need to be performed before one another. In particular, the dependence regression does not consider the shape of the fabricated electrode. The results on stress and deformation calculation are only mentioned in the conclusion, regarding electrode tool mounting. One can assume that it also helps optimizing the fabrication process, but information lack on such results. On the other hand, a complete description of the model and mathermatical basis for stress simulation is given, but the only conclusion from the resulting values of deformation (figure 7) is that displacement is greater at electrodes corners. No results are given concerning circular electrodes.
The suggestion is the to change the paper structure to better show the complementarity of each part
Detailed comment about the structure and additional remarks are given below.
Technical detail
Structure of the article
The paper is organized in four parts: introduction, materials and methods, results, and conclusion. However, the content of each part seem not always to fit well with this organization. The description of the model used for "simulation of internal stresses" (part 3.1) should be placed in part 2 (materials and methods). Only the last part (from page 10, line 261 "To perform the analysis..." to the end of part 3.1 should be kept in the "result" part. Besides, parts 3.2 and 3.3 deal with two different topics: part 3.2 presents the results of regression dependence for general EDM operating parameters, while part 3.3 gives the results obtained with the fabricated electrode tool. These two parts could be better organized.
According to what authors want to put forward, the paper structure should be revised. Suggestions could be:
- Present one part dedicated to general EDM study (regression dependence) and one part on electrode tool fabrication and testing.
- Better show the complementarity of the two parts, for instance by studying the effect of discharge parameters (U, I, Ton) for the two electrode-tool configurations (square and cylindrical).
- Split the paper in two articles, on considering EDM input parameters influence and one dealing with fabrication of electrode tools (including modelling of stress and displacement, with deeper explanation on how the results will help improve the fabrication).
Oher minor issues
-
The output of the regression dependence is given as the value of the EDM cut width (page 69, line 131/132). However, in the vertical axis label in figures 8 to 10 the output Q is expressed in mm/h, which would correspond to a cutting speed (presumably in terms of width cutting speed). This unit is also used in the text below Figure 8, but for the text describing figures 9 and 10 it changes back to units in mm. It should be clarified if cutting with or speed is considered, and only one notation should be chosen.
-
Page 3, line 81: what does mean the expression "laser intensity was 2000 mA"? Does this refer to the electrical current input of the laser power supply, or to the laser light intensity (in that case the unit "mA" is not appropriate)? Since the power of the laser is already given (50 W) this information might be omitted (in addition, the laser is included in the SLM-50 3-D printer and the setting of current is expected to be specifically related to this device. Laser power seems to be a most relevant information)
-
Check equation 15: it seems that some Cyrillic letters "ILI" remain on the left term.
-
Page 9, line 246: This sentence looks like a subsection title. This could be better presented.
Clarity of expression
The paper is generally well written but some improvements could be made:
-
The first letter is missing (The study and developpment...) in the abstract (page 1, line 9)
-
The use of abbreviations without a first explanation should be avoided in the abstract:
-
The abbreviation "ET" should be explained before its first used in the abstract: line 12, "Modeling of the electrode-tool (ET)". In addition, the term "tool electrode" is frequently used (including in the title). It should be better to keep always the same expression (choose between electrode-tool, electrode tool or tool electrode).
-
The abbreviation "SLM" (selective laser melting) should be explained when first use. It is only written fully at page 2, line 64
-
-
Page 2, line 51: A capital letter is missing after the full stop (Most of the metal...)
Presentation
Diagrams, tables and captions
-
Figure 2: the scale on the figure should be indicated as "50 µm" instead of 50 MKM"
-
The meaning of the red pattern between the 70 x 40 mm working space cylinder and the electrode tools in Figure 5 c is not clear. Additional comments should be given.
-
Figure 6 adds little information, since the visual rendering of the ET model with ANSYS Additive workspace is similar to that of Materialize Magics (except for the color from bronze to red). A simple sentence could express that. Otherwise the purpose of this figure should be more clearly justified.

Author Response
Dear Reviewer,
I am grateful for the helpful and interesting comments by you.
We have revised the article.
We tried to make it better.
With best regards!
Timur Rizovich Ablyaz

Reviewer 2 Report
The work entitled "Investigation of the texturing process by electro-discharge machining using the technology of creating tool electrodes by selective laser melting" has been presented for review.
1. I recommend shortening the title.
2. Abstract - please check the typos - line 9.
3. From the abstract it is not clear what you mean by "ET", please specify the abbreviations.
4. lines 20-21 "The fundamentals of the technology 20
for creating a textured surface of the ET produced by the SLM method have been developed." - What "fundamentals" were developed?
5. Many of the reported statements in the introduction require relevant references: e.g., lines 36-38; 55-57;
6. lines 63-65 - this sentence is too long, please rephrase.
7. Moreover, to understand the issue of why the manufacturing of ET is possibly challenging, you first of all need to provide some explanation about the product. You said that ET is made of copper, brass, bronze, etc. Ok, all of these metals are printable by SLM, so what was the gap to print ET? The geometry? the requirements to surface? It should be explained very clearly.
8. lines 65-70 "The development of EDM technology by grown ET obtained by the SLM method for applying a textured surface is relevant [15–16] (Figure 1)" It is really hard to understand what you talking about. Moreover, what actually we see in Fig.1? Please, provide a clear description of the current gaps and what research questions you tried to solve.
9. lines 71-74 "The study and development of the technological foundations for creating a textured surface using an electrode - a tool obtained by the method of additive manufacturing, is the purpose of the work." - The same here - the novelty of the research should be based on the recent publications. What has been done before, and what has not been done but you believe the industry/academy needs that?
10. Fig. 2 - the scale bar should be in accordance to SI, microns - μm. The capture is also confusing, what do you mean by "form"? The powder morphology? The microscopy method also should be defined.
11. In Chapter 2, the PSD data of the powder is missing, and its chemical composition, supplier, etc.
12. Line 86 - what experiments do you mean here? Again, the methods and materials are described in a bit confusing manner - you started with the printer and powder, ok, then, how many samples and of what size/geometry were produced?
And then what tests were applied to all groups of the samples.
13. Figs. 3-5 and others - scale bar is needed here.
14. 111-113 - in metal SLM process supports and the component always are made from the same material.
15. Line 263 - check and correct the figures numbering
16. lines 309-310 "ET with a square base, - ET with a round base are obtained as a result of modeling and manufacturing using the SLM technology (Figure 11)" - it is also rather confusing since the geometry is rather affordably by SLM, and the used powder is also well-known and its printing parameters were reported in multiple publications. So was your modeling really necessary for printing of these samples?
17. The sections of results and conclusion need to be rewritten.
Again what was the main goal of your research? To make a predictable model of SLM of Ti64 of specific components?
Or to print and test the properties of the Ti64 component?
Or to compare the traditionally manufactured ET part with the additively manufactured one?
The paper has serious flaws which need to be fixed.
18. Some recent papers that can help to improve your manuscript:
https://doi.org/10.1016/j.precisioneng.2021.01.002
https://doi.org/10.3390/ma14164473
https://doi.org/10.3390/jmmp5040106
Author Response

(The authors gave the same response as above.)

Round 2
Reviewer 1 Report
The requested modifications have been made. With the choice of focusing on the electrodes (and keeping the part on regression dependence for another article) the article structure is more clear.
The other minor comments have also been properly dealt with. The paper is now suitable for publication.
Reviewer 2 Report
The authors have answered all my comments to some extent. I still think that all the sections could be improved by the more logical presentation and clear motivation of the research but the paper can be recommended for publication even in its current form.